# Mobility data improve forecasting of COVID-19 incidence trends using Graph Neural Networks (Extended Abstract)

Simon Witzke
simon.witzke@hpi.de
Hasso Plattner Institute, Digital Engineering Faculty,
University of Potsdam

Noel Danz
noel.danz@hpi.de
Hasso Plattner Institute, Digital Engineering Faculty,
University of Potsdam

Katharina Baum
katharina.baum@hpi.de
Department of Mathematics and Computer Science, Free
University Berlin
Hasso Plattner Institute, Digital Engineering Faculty,
University of Potsdam

Bernhard Y. Renard
bernhard.renard@hpi.de
Hasso Plattner Institute, Digital Engineering Faculty,
University of Potsdam

## ABSTRACT

The COVID-19 pandemic has had a considerable global impact over the last few years. Many efforts were made to understand and estimate its development. The availability of large amounts of data, including mobility data, has led to numerous Graph Neural Networks (GNN) being proposed to leverage this data and forecast case numbers for the short-term future. However, information about trend developments, especially where trends reverse directions, is crucial in informing decisions. GNNs may be able to use information from regions where trends change first to improve predictions for locations with delays. We consider the first omicron wave in Germany at the end of 2021 and compare a heterogeneous GNN using mobility data with a model without spatial information. We observe that, for this period, mobility data significantly improve forecasts and specifically that improvements occur earlier in time. Using GNNs and mobility data enables leveraging information from counties affected earlier to improve forecasts for counties affected later. We conclude that such performance improvements could be transferred to counties with earlier change points by also including neighboring nations in the graph structure. Further, we emphasize the need for systematic contextual evaluation of GNN-based models for forecasting pandemic trends.

## KEYWORDS

mobility data, trend estimation, graph neural networks, covid-19

**ACM Reference Format:**
Simon Witzke, Noel Danz, Katharina Baum, and Bernhard Y. Renard. 2023. Mobility data improve forecasting of COVID-19 incidence trends using Graph Neural Networks (Extended Abstract). In *epiDAMIK 2023: 6th epi-DAMIK ACM SIGKDD International Workshop on Epidemiology meets Data Mining and Knowledge Discovery, August 7, 2023, Long Beach, CA, USA.* ACM, New York, NY, USA, 5 pages.

## 1 INTRODUCTION

Spreading from Wuhan, China, in late 2019, the COVID-19 pandemic has held humanity in its grasp until recently [35]. The pandemic has had drastic consequences, with estimates of almost fifteen million excess deaths only in 2020 and 2021 [20] and considerable economic and social damages [5]. The global scale of the pandemic led to large amounts of data on different modalities related to epidemic spread being shared, such as mobility and sequencing data. These have been made available to support the development of forecasting methods intended to inform decision makers concerning potential interventions [21, 23]. Human mobility is a central driver in the geographical spread of epidemics caused by air-borne diseases [3], enabling the virus to travel between regions and, in the case of COVID-19, rapidly infecting most of the world. During the pandemic, researchers have combined mobility networks with mechanistic models to understand the influences of changed mobility behavior and further highlight its importance for the pandemic's development [4, 30]. Schlosser et al.[30] have shown that lockdowns strongly impacted mobility structures during the first COVID-19 wave in Germany and that the associated reduction in mobility can slow the virus' geographical spread.

Various spatio-temporal approaches using Recurrent Neural Networks and EXtreme Gradient Boosting have been proposed to forecast county-level COVID-19 metrics [11, 18, 22, 34]. However, recent advances in deep graph learning have led to Graph Neural Networks (GNNs) gaining popularity in domains as diverse as traffic forecasting [12] or computational chemistry [26]. Human mobility between geographical regions can naturally be represented as graphs, where nodes represent locations, such as counties, and edges movements between them. Consequently, numerous approaches that try leveraging the power of GNNs to forecast COVID-19-related metrics, such as cases, deaths, and hospitalizations, have been proposed [9, 10, 13, 24]. These approaches have shown promising results in providing insights into the short-term development of the COVID-19 pandemic. However, informing decision makers about a trend forecast rather than exact numbers might be more beneficial. Communicating trends can be easier than directly communicating cases or deaths. Trends are strong indicators of relevant changes in the

pandemic development and a need for interventions, and their interpretation is straightforward. For example, the US Government used a 14-day downward trend in COVID-19 cases as a condition for potential re-openings [6]. For this purpose, systematically evaluating GNN-based methods' ability to correctly forecast trends is essential. Accurate forecasts are especially relevant for phases with change points, where locations successively experience a change in their trend, such as the peak of a wave.

There are secondary time series modalities, such as Google search trends and smart body temperature sensors. These modalities potentially reflect changes in trends faster than case numbers. This has been successfully leveraged by Kogan et al.[15] and Stolerman et al.[31] to develop early-warning systems in the United States that detect such trend signals up to weeks in advance. Similarly, GNNs may utilize nodes with leading time series to improve forecasts for nodes with lagging time series by passing information via the underlying graph, i.e., information from locations where changes occur earlier might be beneficial for forecasting locations where similar changes are delayed.

In this work, we investigate whether mobility data can improve forecasts of 14-day linear trends of the COVID-19 incidence. We evaluate county-level forecasts of a heterogeneous GNN for locations experiencing a change point during the second half of the first omicron wave at the end of 2021 in Germany [19], where cases are beginning to decline. We further analyze whether our GNN can utilize information from counties with leading changes for forecasting counties that experience similar changes later. Finally, we discuss the implications for developing and evaluating future GNN-based methods for pandemic forecasting.

## 2 MATERIALS AND METHODS

### 2.1 Graph Construction

Inspired by Kapoor et al.[13], we construct heterogeneous spatiotemporal graph samples with distinct edge types for spatial and temporal connections. We design each graph sample to contain 15 weighted mobility subgraphs, representing movements between the 400 German counties as nodes at successive points in time, $t - 14, ..., t$. We use spatial edges to express these mobility graphs. The directed but unweighted temporal edges then link each county at a time point $t - 14, ..., t$ to its representations on up to seven previous days, connecting the spatial components of the graph. Therefore, each graph sample represents a single point in time while still including historical information from previous days.

We use mobility data [16, 28] to build the spatial edges. The used dataset contains the daily movements of nearly one million mobile phone users in Germany and is non-public due to privacy concerns. The number of mobile phones sending location information varies daily, so we normalize the movements by the daily device count and then re-scale all movements with the average daily device count. We find that the daily mobility networks' adjacency matrices are primarily symmetric, i.e., the opposing edges are highly similar. Therefore, we convert the directed into undirected graphs by summing the weights of the edges in both directions. Finally, we denoise the mobility graphs by removing 30% of the non-zero edges with the lowest edge weights, where edges on the thresholding boundary are removed randomly.

The node features of our graph consist of dynamic and static features. We obtain data on the COVID-19 case numbers starting in January 2020 from the Robert Koch Institute [27] and aggregate the data on the county level, resulting in a total of 400 time series. Countering reporting inaccuracies, we calculate the county-level 7-day incidence, a right-aligned 7-day moving sum normalized by the county population and then scaled by 100,000. Each node at time $t$ has the 7-day incidence of the previous seven days until day $t - 6$ as node features. Additionally, we include a cyclical sine/cosine encoding [33] for the weekday and month. This cyclical encoding aims to improve the learning of short and long-term seasonal effects. Lastly, we use the population density of each county as the only static feature. We collect the census data, such as population size and population density, from the German Federal Office of Statistics [17].

As prediction targets, we use 14-day trends in the COVID-19 incidence obtained from linear approximations. A linear approximation has the advantage that it allows us to estimate the strength of a trend and not only its direction compared to converting the problem to a classification task. For this purpose, we smooth the 7-day incidence time series for the whole dataset to remove remaining artifacts, using a center-aligned 7-day moving average. For each county and time point $t$, we perform a linear regression on this smoothed time series with the known time series values at time points $t + 1, ..., t + 14$ as the dependent variable and the number of days from time $t$ into the future $h \in 1, ..., 14$ as the independent variable. We then use the slope of this regression, representing a linear trend of the COVID-19 incidence over the next 14 days from time point $t$, as the ground truth for our forecasts.

### 2.2 Graph Neural Network

Our GNN is similar to the network used by Kapoor et al.[13] and based on Kipf and Welling's[14] graph convolutional layer. We extend this architecture by using relational graph convolutional layers (R-GCN), an extension for heterogeneous graphs proposed by Schlichtkrull et al.[29] that allows feature updates via multiple edge types, where each edge type has its own set of learned parameters. First, the node features are passed through an initial encoding layer followed by a dropout with a probability of 0.2. Next is a three-layer GNN, each with a dropout probability of 0.5. Like Kapoor et al.[13], we add skip-connections and concatenate the output of the initial encoding layer to the output of each R-GCN layer to preserve local information and counter over-smoothing. Lastly, we use a multi-layer perceptron with a single hidden layer to produce the final prediction. We note that for each graph sample, we only use the embeddings of the most recent spatial subgraph to obtain a single forecast for all 400 counties. All layers have 32 hidden units and use a ReLU as the non-linear activation function, except for the last linear layer, which has 16 hidden units. The output layer uses no activation function, allowing positive and negative trend predictions. We implement our GNN in PyTorch [25] and PyTorch Geometric [7].

### 2.3 Training setup

We use a mean squared error (MSE) regression loss and an ADAM optimizer with a learning rate of 1.33e−4 and weight decay of 1e−5.

We employ a batch size of 128 and train for a maximum of 250 epochs with early stopping, with a patience of 10 epochs without improvement.

We adopt a rolling-origin evaluation approach [32] where we extend the training set by the test sample of the previous iteration. We test from November 10, 2021, until December 19, 2021, with all previous data being used for training and validation. We use all data from January 15, 2020, for training and validation. Therefore, the training and validation set contains 665 samples for the first test sample and grows to 704 samples for the last test sample. Our validation set consists of the day after the last training sample and is used for early stopping and model selection. We always have a 17-day gap between the validation and test samples to avoid information leakage to the test sample while also mimicking a real-world situation where we use all the available data to make a forecast.

To counter the sparseness of training data and avoid conditioning our model too strongly on periods that contain limited information, such as summer periods with low incidences, we oversample the training set by multiplicating specific samples. We combine the global German COVID-19 incidence time series with an exponential function, assigning higher importance to more recent dates. We convert the result into a discrete probability distribution where each sample is assigned a probability. We then draw from this distribution with replacement. We use an oversampling rate of 10.

## 2.4 Evaluation Scenario

While we train our models using an MSE regression loss, this metric is not optimal for evaluating our models' performance. Different counties experience the considered phase of the pandemic differently and a metric dependent on the range of the trend values could bias our evaluation.

Therefore we evaluate the models' performance using the Mean Absolute Percentage Error (MAPE) (Appendix A.1) and the symmetric Mean Absolute Percentage Error (sMAPE) (Appendix A.2). Further, while MAPE and sMAPE provide insight into the error in the magnitude of the trend, we are also interested in the model's ability to predict the direction of the trend. For this purpose, we evaluate our models with an adaption of the Mean Directional Accuracy (MDA) (Appendix A.3).

To investigate if our models can leverage mobility data to improve predictions in counties with lagging change points, we consider the first omicron wave at the end of 2021, from November 10 to December 19. For this period, we extract the date on which each county's corresponding smoothed COVID-19 7-day incidence time series has its maximum, i.e., its peak. We consider this the point when the trend will likely change from positive to negative as the incidence begins to decline.

After obtaining the peak for each county, we use a 7-day moving window to evaluate how the prediction performance develops as more counties reach their peak. For each window, we collect all counties that have their peak inside the current window. We then compute all metrics for these counties using the forecast and ground truth of their peak date and shift the window by one day.

We conduct additional experiments with the same evaluation setup but replace the adjacency matrices of the mobility subgraphs

with identity matrices to verify that difference in performance can be accounted to the mobility data. Thus, we train models with the same number of parameters but do not include spatial information.

## 3 RESULTS

For all experiments, there is a clear performance improvement as more counties reach their peak over time that is consistent across all metrics. This improvement is more pronounced for models with mobility data than those without spatial information (see Figure 1). To verify that our findings that models with mobility data perform better than models without spatial information are significant, we conduct paired one-tail Wilcoxon signed-rank tests with significance level $\alpha = 0.05$ for all metrics. After correcting for multiple testing using the Benjamini-Hochberg method [1], we find that for MAPE (*p-value* $\approx 0.021$), sMAPE (*p-value* $\approx 2.738e-6$), and MDA (*p-value* $\approx 6.661e-6$) the mobility-conditioned models significantly outperform the models without spatial information.

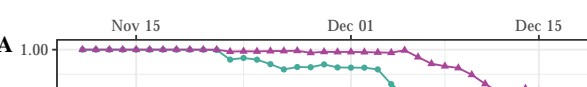

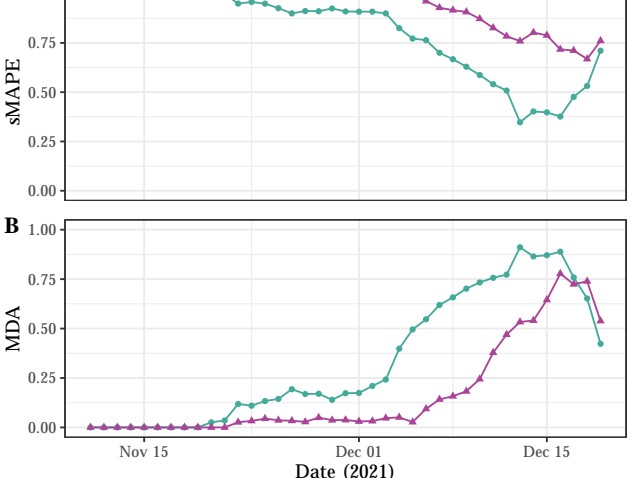

**Figure 1: (A) sMAPE (lower is better) for peaks in 7-day moving windows. The performance improves over time for both experiments before declining. The effect occurs earlier and is greater for models with mobility data. (B) The MDA (higher is better) almost mirrors the sMAPE's behavior. This suggests that while more recent training data improve predictions, this effect is amplified by mobility data.**

Figure 1 (A, B) clearly shows that the improvements in sMAPE and MDA happen earlier and are more extreme for the models with mobility data. This difference indicates that the improvements cannot solely be attributed to the fact that the models have seen more recent and relevant data and are therefore conditioned better. Furthermore, due to the 17-day gap to avoid information leakage, the model is unlikely to have seen any recent negative trends for a county before its peak during training. However, as earlier counties are already past their peak and are experiencing decreasing

incidences, they can share this information with counties where peaks occur later.

## 4 DISCUSSION AND CONCLUSION

We find that mobility data significantly improve forecasting performance compared to experiments without spatial information. We have two hypotheses for our observations. Firstly, the structural information in the mobility networks and their variation over time might lead to improved predictions. Secondly, our GNN model can pick up information from counties that experience changes, such as beginning downtrends in incidences, earlier and use them for forecasts of counties where these changes occur delayed. With our current experimental setup, we are unable to disentangle these hypotheses. However, further experiments, for example, using static spatial connections, could provide insights.

Counties that are the first to experience a change in trend seem unable to benefit from mobility data. However, these counties might be of the highest interest as changes occur earlier and are likely more vital indicators of the need for interventions. Therefore it could be valuable to include additional nodes representing neighboring nations in our graph to leverage potentially leading information from them.

Our analysis suggests that systematically analyzing models' capabilities of making accurate trend forecasts during times of interest is highly valuable. Different components, such as the magnitude and direction of a trend, are relevant for providing a holistic understanding in an epidemiological context. It could be helpful to extend evaluations by applying post-hoc explainability methods for graph-based models to understand better how the models make their predictions. Such explanations could provide insights for epidemiologists to construct hypotheses regarding the pandemic's current state and spreading behavior.

We showed the capabilities of a heterogeneous spatio-temporal GNN in leveraging mobility data to improve forecasts for counties with lagging time series directly after a change in trend. We suggest that including more global information via nodes representing other nations could extend this effect to leading counties where changes occur first. Currently, we evaluate single rolling-origin evaluation experiments for the change point of the COVID-19 pandemic in Germany. To substantiate our findings, we will consider different phases of the pandemic, including change points with a switch to upward trends. Furthermore, we will run experiments repeatedly to verify the robustness of our results and establish confidence bounds.

## ACKNOWLEDGMENTS

This work was supported by the German BMWK through the DAKI-FWS project [01MK21009E to B.Y.R.].

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

## A   EVALUATION METRICS

### A.1   Mean Absolute Percentage Error

The Mean Absolute Percentage Error (MAPE) [8]:

$$MAPE = \frac{1}{n} \sum_{i=1}^{n} \frac{|\hat{y}_i - y_i|}{|y_i|},$$

where $n$ is the number of counties, is a relative error independent of the range of values. The MAPE is highly susceptible to observations close to zero causing the metric to explode. A smaller MAPE is better.

### A.2   symmetric Mean Absolute Percentage Error

The symmetric Mean Absolute Percentage Error (sMAPE) [8]:

$$sMAPE = \frac{1}{n} \sum_{i=1}^{n} \frac{|\hat{y}_i - y_i|}{|\hat{y}_i| + |y_i|},$$

where $n$ is the number of counties, is another relative metric that takes on values between 0 and 1 and is therefore protected against exploding values. A smaller sMAPE is better.

### A.3   Mean Directional Accuracy

We use an adaption of the Mean Directional Accuracy (MDA) [2]. As we only forecast a single value, the MDA can be simplified, yielding the rate at which the models can identify the trend correctly:

$$MDA = \frac{1}{n} \sum_{i=1}^{n} \mathbf{1}_{sign(\hat{y}_i)=sign(y_i)},$$

where $n$ is the number of counties and $\mathbf{1}$ the indicator function. A larger MDA is better.