# OpenReview forum: "Mobility data improve forecasting of COVID-19 incidence trends using Graph Neural Networks (Extended Abstract)"
_KDD.org/2023/Workshop/epiDAMIK — KDD 2023 Workshop epiDAMIK_

### Official Review · Reviewer_w1yT · 2023-06-28
**Mobility data improve forecasting of COVID-19 incidence using Graph Neural Networks**

**Rating:** 3
**Confidence:** 3

**Review:**


Quality
The paper is written in an understandable manner. The experiments are done on (or shown on) a limited set and the authors don’t compare with other baseline methods for case trend analysis of COVID-19

Clarity
It can be improved further to make contributions of the authors clearly delineated from the existing literature. The explanation of the graph can be better if it is shown visually, but I understand that there are space limitations for this submission

Originality
The idea is something that is already explored by other researchers to answer different or similar questions related to the pandemic forecasting

Significance
The work is significant to the workshop given the problem it is tackling. This is also interesting to society given the possibilities of other epidemics

Pros
Compare different error metrics and also perform a statistical test to show the significance
They show that the addition of mobility is helping in the forecasting; counties with earlier trends can also help predict those with later.

Cons
No baseline comparison apart from their own method without spatial information
Missing citations:
1) mobility network related - Mobility network models of COVID-19 explain inequities and inform reopening; this and follow-up works seem closely connected to what the authors explore
2) Trends are more important than actual numbers in terms of pandemic forecasts
3) Using information from one county to help the other
I am aware of works from the CDC forecasting hub, XPRIZE pandemic challenge, etc that have discussed these issues. It would be relevant to cite those papers and show how this work is different

---

> ### Author Response · Authors · 2023-07-28
> **Further baseline comparisons will be part of future work**
>
> We thank the reviewer for their helpful comments. We agree that comparing our method to additional baselines apart from our model without spatial information would be helpful. However, our focus for this manuscript was on the role of mobility data; therefore, such a comparison will be left to future work.
> We agree that a visual representation of the graph would be beneficial but decided against it for the manuscript due to space restrictions. However, we will show such a visualization in our presentation and poster.
>
> We appreciate the reviewer pointing out potentially relevant references. While “Mobility network models of COVID-19 explain inequities and inform reopening” (Chang et al., 2021) uses mobility data obtained from mobile phones, their methods differ considerably from ours as they use mechanistic models rather than graph neural networks. We extended our introduction to highlight their work’s relevance in showing the impact of mobility data on the pandemic.
> > “During the pandemic, researchers have combined mobility networks with mechanistic models to understand the influences of changed mobility behavior and further highlight its importance for the pandemic's development. (Chang et al., 2021, Schlosser et al., 2020)”
>
> We extended our argument on why forecasting trends might be more beneficial than directly forecasting actual numbers:
> > “However, informing decision makers about a trend forecast rather than exact numbers might be more beneficial. Communicating trends can be easier than directly communicating cases or deaths. Trends are strong indicators of relevant changes in the pandemic development and a need for interventions, and their interpretation is straightforward. For example, the US Government used a 14-day downward trend in COVID-19 cases as a condition for potential re-openings (Duffey and Zio, 2020).”
>
> We added references to various papers that use different spatio-temporal approaches to forecast relevant metrics:
> > “Various spatio-temporal approaches using Recurrent Neural Networks and EXtreme Gradient Boosting have been proposed to forecast county-level COVID-19 metrics (Hssayeni et al., 2021, Lucas et al., 2023, Nikparvar et al., 2021, Vahedi et al., 2021).”

---

### Official Review · Reviewer_8yuZ · 2023-06-28
**A Comprehensive Review of a COVID-19 Forecasting Paper: Promising Approach with Room for Clarification and Improvement**

**Rating:** 3
**Confidence:** 4

**Review:**

Overall, the paper is well-written and provides valuable insights into the potential benefits of using mobility data and GNNs for COVID-19 forecasting. While the paper's quality is commendable, with a clear research objective and a well-defined methodology, the evaluation is lacking and somethings need better explaining (detailed in the list below). The paper's clarity is also good but needs improvement in a few areas. The significance of this work is evident in its potential to improve pandemic forecasting and inform decision-making but the originality of leveraging mobility data to enhance predictions is not something new.


Pros:

Model architecture and training setup: The paper mentions that the model architecture is well chosen and the training setup and evaluation metrics are appropriate. These aspects are positively reviewed.

Preprocessing: The paper's preprocessing steps of smoothing the 7-day incidence data using a center-aligned 7-day moving average (and normalizing where necessary) is considered a good approach to enhance the quality of the data and facilitate more accurate trend analysis.

Cons:
Accuracy of predicted trends: Using linear regression to predict the 14-day trends raises questions about the accuracy of these predicted trends themselves. It might be more appropriate to linearly approximate the ground truth data when evaluating the model's performance since that is the target being predicted.

Evaluation scenario: The evaluation scenario seems to be lacking since it only considers a one-month period. There is confusion about how the peak is determined using a 7-day moving window. If the window is limited to 7 days, there will always be a peak within that period since the paper mentions "We extract the date on which each county’s corresponding smoothed COVID-19 7-day incidence time series
has its maximum, i.e., its peak". If it is simply the peak of ALL 7day incidence time series for each county then it is more than reasonable, if not then further clarification is needed to understand the evaluation process.

Training data size: The paper does not clearly state the size of the training data. It mentions considering the 2nd half of the first omicron wave from November 10 to December 19, which implies two months of data. However, it is unclear when the summer period is included in the training data.

Oversampling: The necessity of oversampling is questioned, especially when predicting 14-day trends since low incidence should not have effect on trends. It is also hard to determine the negative effects of this since the training data's size is not mentioned. If ALL previous periods are used for training, there is a concern that discrete oversampling may not be suitable as different waves of variants can behave differently due to varying growth rates. While if only the brief period (1st half of the Omicron wave is used), it can potentially lead to overfitting due to a small amount of data, this is especially the case when the model is evaluated during a wave of a different variant.

Testing on declining trends only: It is questioned why the evaluation is conducted only on a period where the trend is declining. The evaluation should consider periods of both increasing and decreasing trends to provide a comprehensive assessment of the model's performance.



In conclusion, while the paper presents some positive aspects, including the model architecture, training setup, and evaluation metrics, there are significant concerns and areas that require clarification and justification. The questionable use of predicted trends as target data, the unclear evaluation scenario and peak determination, and the potential issues with oversampling can raise some doubts about the methodology. It is also not clearly discussed how the GNN can utilize information from counties with leading changes for forecasting
counties that experience similar changes later. For future work, it would also be good to provide comparisons to other non GNN related models (even something as simple as the linear regression being done). Further elaboration and addressing of these concerns would greatly improve the quality and clarity of this work. The title of the paper can also potentially be changed since the forecasting being done is of COVID-19 trends rather than incidence.

---

> ### Author Response · Authors · 2023-07-28
> **Extended experiments with additional baselines and pandemic phases and empirical justification of oversampling are planned**
>
> We appreciate the reviewer’s detailed comments. We understand the reviewer's concern regarding the accuracy of the predicted trends. We have clarified our description further as our current approach already follows the reviewer's suggestions. We extended the description of our target generation to show that we approximate linear trends on the known ground truth data and, therefore, an accurate linear approximation of the original smoothed incidence time series. Nevertheless, we slightly adjusted the manuscript to communicate this more clearly. This paragraph now reads:
> > “As prediction targets, we use 14-day trends in the COVID-19 incidence obtained from linear approximations. A linear approximation has the advantage that it allows us to estimate the strength of a trend and not only its direction compared to converting the problem to a classification task. For this purpose, we smooth the 7-day incidence time series for the whole dataset to remove remaining artifacts, using a center-aligned 7-day moving average. For each county and time point t, we perform a linear regression on this smoothed time series with the known time series values at time points t+1,...,t+14 as the dependent variable and the number of days from time t into the future h in {1,...,14} as the independent variable. We then use the slope of this regression, representing a linear trend of the COVID-19 incidence over the next 14 days from time point t, as the ground truth for our forecasts.”
>
> We agree that extending the evaluation period to more than a one-month period would be beneficial and will be addressed in future work. Concerning the confusion of the peak estimation and the 7-day window, we added further clarification to the manuscript. We do use the whole one-month period to find the peak for each ground truth smoothed 7-day incidence time series, i.e., each county. However, when we calculate the evaluation metric, we only use those counties that have their peaks in a 7-day moving window with a step size of one. This means that all counties that have their peak within a window contribute to the metric with the forecast made for their peak date.
> > “For this period, we extract the date on which each county's corresponding smoothed COVID-19 7-day incidence time series has its maximum, i.e., its peak. We consider this the point when the trend will likely change from positive to negative as the incidence begins to decline.
> > After obtaining the peak for each county, we use a 7-day moving window to evaluate how the prediction performance develops as more counties reach their peak. For each window, we collect all counties that have their peak inside the current window. We then compute all metrics for these counties using the forecast and ground truth of their peak date and shift the window by one day.”
>
> Furthermore, we clarified the exact size of the training/validation dataset:
> > “We use all data from January 15, 2020, for training and validation. Therefore, the training and validation set contains 665 samples for the first test sample and grows to 704 samples for the last test sample.”
>
> We understand the reviewer’s concern that discrete oversampling might be unsuitable due to varying growth rates during different waves. We will provide extensive empirical justification for oversampling in future work. Nonetheless, we have extended our description and provided further details of the oversampling processes to be more comprehensive.
> > “We combine the global German COVID-19 incidence time series with an exponential function, assigning higher importance to more recent dates. We convert the result into a discrete probability distribution where each sample is assigned a probability.”
>
> We agree with the reviewer’s comment that evaluating upward and downward trends would be good. However, extracting change points for upwards trends is more complex and was therefore out of scope for this work. We will extend to additional evaluation periods for future work, including changes from neutral and negative trends to upwards trends.
>
> Finally, the reviewer’s suggestion of extending the title of our manuscript to include “trend” is very valuable and we have updated the title accordingly. Furthermore, we added a clarification that our manuscript is an extended abstract.

---

### Official Review · Reviewer_y98U · 2023-06-30
**Mobility data added to GNN architecture to improve outbreak forecasting**

**Rating:** 3
**Confidence:** 3

**Review:**

### Summary
This paper seeks to improve graph neural network driven epidemic forecasting by incorporating mobility data. They focus on forecasting the reversal of trends using this additional information. The effect of the mobility data is isolated by comparing the results with a similar GNN without mobility data. They emphasize the value that mobility data holds in this application compared to other sources of data. Although their results show promise for limited settings, they do not consider settings in which the trends changes in direction from downwards to upwards, which is arguably more relevant for disease forecasting.

### Strengths
- The authors test under real-world assumptions of data availability lags by using a 17-day gap between their training and testing sets.
- The contribution is simple and clearly stated, as well and the hypotheses they introduce concerning their results.

### Weaknesses
- Limited to a linear trend of incidence over 14 days. It is unclear whether this means the method only has the capability to detect changes in trend.
- They only evaluate an upwards to downwards trend, which may affect the viability of the method until the opposite is verified.

### Suggestions
- If possible, it would help to have a small GNN architecture diagram
- When you “design each graph sample to contain 15 temporal connections…”, does each “graph sample” represent a single time point?
- Elaborate more on the purpose of the cyclical sine/cosine encoding

---

> ### Author Response · Authors · 2023-07-28
> **Additional experiments with periods with a switch to upward trends are following in future work**
>
> We thank the reviewer for their comments. We agree with the reviewer that testing not only on trend changes from upwards to downwards but also vice versa is essential. In our opinion, detecting a change from upwards to downwards can be valuable in informing about the possibility of lifting restrictions. However, correctly identifying such change points in the incidence time series for evaluation is more complex than selecting a peak as an indicator for a trend change from upward to downward and was therefore outside the scope of this work. We will, however, include this in future work. We do not see the limitation to a linear 14-day trend and the restriction to detecting trend changes as a weakness. However, we need to set a specific horizon for the trend and believe that 14 days is reasonable. Our method estimates a trend's direction and strength, providing decision makers with valuable information while removing the complexity of predicting and interpreting exact case numbers.
>
> We agree that an architecture diagram of the GNN would aid in understanding our model. We will show such a diagram during our presentation and poster. To make our manuscript better understandable, we have clarified that each described graph sample indeed represents a single point in time:
> > “Therefore, each graph sample represents a single point in time while still including historical information from previous days.”
>
> Similarly, we have expanded on the purpose of the cyclical sine/cosine encoding:
> > “Additionally, we include a cyclical sine/cosine encoding (Taylor and Letham, 2018) for the weekday and month. This cyclical encoding aims to improve the learning of short and long-term seasonal effects.”